# Preparation of S-C_3_N_4_/AgCdS Z-Scheme Heterojunction Photocatalyst and Its Effectively Improved Photocatalytic Performance

**DOI:** 10.3390/molecules29091931

**Published:** 2024-04-24

**Authors:** Yuhong Lin, Zhuoyuan Chen, Chang Feng, Li Ma, Jiangping Jing, Jian Hou, Likun Xu, Mingxian Sun, Dongchu Chen

**Affiliations:** 1School of Materials Science and Hydrogen Energy, Foshan University, 18 Jiangwanyi Road, Foshan 528000, China; lyh6799566@163.com (Y.L.); jpjing@fosu.edu.cn (J.J.); chendc@fosu.edu.cn (D.C.); 2State Key Laboratory for Marine Corrosion and Protection, Luoyang Ship Material Research Institute (LSMRI), Wenhai Road, Qingdao 266237, China; mal@sunrui.net (L.M.); houjian725@126.com (J.H.); xulk@sunrui.net (L.X.); sunmx@sunrui.net (M.S.); 3Guangdong Key Laboratory for Hydrogen Energy Technologies, 18 Jiangwanyi Road, Foshan 528000, China

**Keywords:** semiconductor photocatalysis, Z-scheme heterojunction, S-doped graphitic carbon nitride, Ag-doped CdS nanoparticles, photocatalytic degradation

## Abstract

In this study, S-doped graphitic carbon nitride (S-C_3_N_4_) was prepared using the high-temperature polymerization method, and then S-C_3_N_4_/AgCdS heterojunction photocatalyst was obtained using the chemical deposition method through loading Ag-doped CdS nanoparticles (AgCdS NPs) on the surface of S-C_3_N_4_. Experimental results show that the AgCdS NPs were evenly dispersed on the surface of S-C_3_N_4_, indicating that a good heterojunction structure was formed. Compared to S-C_3_N_4_, CdS, AgCdS and S-C_3_N_4_/CdS, the photocatalytic performance of S-C_3_N_4_/AgCdS has been significantly improved, and exhibits excellent photocatalytic degradation performance of Rhodamine B and methyl orange. The doping of Ag in collaboration with the construction of a Z-scheme heterojunction system promoted the effective separation and transport of the photogenerated carriers in S-C_3_N_4_/AgCdS, significantly accelerated its photocatalytic reaction process, and thus improved its photocatalytic performance.

## 1. Introduction

Solar energy is inexhaustible and it is one of the most abundant energy sources that people can obtain on Earth [1]. In order to alleviate global energy shortages and mitigate environmental pollution, photocatalytic technology that is in line with the sustainable development of human society has emerged [2,3,4]. At present, the degradation of organic pollutants in water or air under artificial or natural light irradiation has been widely studied [5]. Although photocatalytic technology has become a hot topic in recent years, it must be acknowledged that the recombination speed of the electron–hole pair generated by single component photocatalysts is very fast, and only 10% of the separated charges can successfully reach the surface of the photocatalyst to undergo chemical reactions [6]. This is an extremely low utilization of solar light energy, that is, the light conversion efficiency in the light reaction is very low. In order to improve and overcome these problems, reasonable modification of photocatalysts has become necessary. Morphological control, elemental doping, and the construction of heterojunction systems are effective ways to improve photocatalytic performance [7,8,9], playing a very important role in alleviating current environmental pollution and energy crisis issues.

Graphitic carbon nitride (g-C_3_N_4_) and CdS are two hot materials in the field of photocatalysis [10,11]. g-C_3_N_4_ has a two-dimensional graphite-like phase structure, which can provide a large number of active sites in the photocatalytic reaction process and is conducive to the rapid progress of the photocatalytic process [12]. Narrow-band gap CdS material (2.4 eV) can effectively absorb and utilize visible light, and it has a fast electron transmission rate and good photocatalytic performance [13]. However, the recombination of the photoinduced charge carriers generated by individual photocatalysts of g-C_3_N_4_ or CdS is fast, which hinders their effective reactions in the photocatalytic process and limits their further application in the field of photocatalysis. By establishing an effective heterojunction system, the recombination of photoinduced charge carriers can be effectively suppressed. Under the action of heterojunction electric fields, the photogenerated charge carriers can be effectively separated, and their photocatalytic performance can be significantly improved. Xu et al. [14] found that the CdS/g-C_3_N_4_ composite photocatalyst can significantly shorten the migration distance of the photogenerated carriers to the interface, realize the rapid separation of photogenerated electron–hole pairs, and reduce the recombination of photogenerated carriers. The degradation rate of azo-dye methyl orange (MO) by composite CdS/g-C_3_N_4_ is 93.8%, and its photocatalytic activity is much better than that of CdS and g-C_3_N_4_ alone. Xu Y et al. [15] prepared CdS/g-C_3_N_4_ for photocatalytic degradation of Rhodamine B (RhB), and found that ⋅O^2−^ was the major oxidation species for photocatalytic degradation. They proved that the photocatalytic activity of CdS/g-C_3_N_4_ was significantly enhanced compared with that of single-component photocatalysts. Li et al. [16] successfully prepared g-C_3_N_4_/CdS composites with a Z-scheme structure, and found that the Z-scheme heterojunction structure could inhibit the photoinduced self-corrosion of CdS, and verified that its photocatalytic activity on erythromycin (Ery) and tetracycline (TC) was significantly enhanced compared to that of pure g-C_3_N_4_ and pure CdS. Therefore, the rational design of an effective heterojunction photocatalytic system is of great significance for improving the photocatalytic performance of materials.

Doping is another effective way to improve the performance of the photocatalyst. Doping is the process of introducing foreign elements into the parent photocatalyst without generating new crystalline forms, phases, or lattice structures, and its purpose is to adjust the energy band structure of the photocatalyst by a very small amount, so as to make more efficient use of visible light and improve the separation efficiency of photogenerated electron–hole pairs [17]. Since Liu et al. [18] verified that sulfur-doped carbon nitride (S-C_3_N_4_) has efficient photocatalytic performance in 2010, a large number of subsequent corresponding studies has been carried out based on S-C_3_N_4_ [19,20], and photocatalysts with excellent performance have been obtained. At the same time, many doped photocatalysts, such as doped TiO_2_ [21], doped ZnO [22], doped CdS [23], doped WO_3_ [24], etc., have been widely considered because of their good photocatalytic performance. Establishing an effective heterojunction system with doped photocatalysts is bound to further enhance the photocatalytic performance of the photocatalysts.

Based on this, we first prepared S-C_3_N_4_, and then constructed effective S-C_3_N_4_/AgCdS heterojunction by further loading Ag-doped cadmium sulfide nanoparticles (AgCdS NPs) onto the surface of S-C_3_N_4_ through chemical deposition. The photocatalytic performance of the prepared photocatalysts is studied in the present paper. The S-C_3_N_4_/AgCdS photocatalyst possesses excellent photocatalytic performance in the degradation of RhB and MO due to the synergistic effect of Ag doping and the constructed Z-scheme heterojunction. Compared to type-II heterojunction systems, Z-scheme heterojunctions have stronger redox capabilities to promote rapid removal of pollutants. This work utilized doped photocatalysts to establish an effective heterojunction system and significantly improved its photocatalytic performance. In the process of establishing heterojunctions, various modification methods of photocatalysts such as doping, morphology control, and heterojunction establishment were fully utilized. This provides a good reference for designing effective photocatalytic systems.

## 2. Results

The XRD patterns of the prepared photocatalysts are shown in Figure 1a. The diffraction peaks of CdS are observed at 26.5°, 43.8°, and 51.9°, which correspond to the (111), (220), and (311) crystal faces of the face-centered cubic structure CdS, respectively, and are consistent with the standard card JCPDS 65-2887 [25]. Figure 1b shows the XRD patterns near the characteristic diffraction peak at 2θ of 26.5°. The locally magnified spectra near 26.5° show a slight left shift in the characteristic diffraction peak of AgCdS compared to that of the (111) crystal plane of CdS, indicating the successful doping of Ag into the CdS lattice structure. For S-C_3_N_4_, the strongest diffraction peak at 27.7° is assigned to the (002) diffraction plane reflected by the interlayer stacking of aromatic structure [26,27,28]. This matches well with the standard card JCPDS 87-1526 [29]. For the XRD patterns of S-C_3_N_4_/CdS, the diffraction peaks are consistent with those of S-C_3_N_4_ and CdS, respectively. Similarly, for the XRD patterns of the S-C_3_N_4_/AgCdS photocatalyst, the characteristic diffraction peaks of S-C_3_N_4_ and AgCdS can be clearly observed. The XRD results in Figure 1 indicate that the S-C_3_N_4_/CdS and S-C_3_N_4_/AgCdS photocatalysts are successfully prepared.

Figure 2 shows the microstructure and elemental mapping of the prepared photocatalysts. Figure 2a shows the SEM image of S-C_3_N_4_, and the layered stacking structure can be clearly observed. The CdS NPs shown in Figure 2b exhibit a spherical cluster structure, which may have transformed into clusters assembled by nanoparticles due to water-bath heating and aggregation. Figure 2c shows the SEM image of the prepared AgCdS nanoparticles, which exhibit the same morphology and structure as CdS nanoparticles shown in Figure 2b. Combined with the XRD results shown in Figure 1 and the SEM results shown in Figure 2b,c, Ag doping does not change the crystal and morphology of CdS. Figure 2d,e are the SEM images of S-C_3_N_4_/CdS and S-C_3_N_4_/AgCdS, respectively, and the lamellar structure of S-C_3_N_4_ can still be observed. In addition, the nanoparticle structure can be observed on the lamellar surface of S-C_3_N_4_, which is attributed to the deposition of CdS and AgCdS nanoparticles. By further enlarging the SEM image of S-C_3_N_4_/AgCdS, as shown in Figure 2f, it can be clearly observed that nanoparticles are uniformly deposited on the surface of S-C_3_N_4_ nanosheets. Figure 2g,h are the TEM images of the S-C_3_N_4_/AgCdS photocatalyst, where the deposition of nanoparticles onto nanosheets can be observed. By further observing the HRTEM images (Figure 2h), the S-C_3_N_4_ edges (the red dashed line area) of sheet-like structures and lattice stripes of AgCdS nanoparticles can be observed. The lattice fringes with spacing of 0.17, 0.21, and 0.33 nm correspond to the (311), (220), and (111) crystal faces of AgCdS. Figure 2i shows the scanning elemental distribution of S-C_3_N_4_/AgCdS, in which the presence of C, N, Ag, Cd, and S elements can be observed. Each element is evenly dispersed in the scanning area and fully matches with the morphology and structure obtained, indicating that AgCdS are uniformly distributed on the surface of S-C_3_N_4_. The results shown in Figure 1 and Figure 2 demonstrate that, for the prepared S-C_3_N_4_/AgCdS photocatalyst, AgCdS nanoparticles are uniformly deposited on the surface of S-C_3_N_4_ nanosheets and form a good heterojunction structure.

XPS was used to further explore the elemental state and valence bond binding information of the prepared S-C_3_N_4_/AgCdS photocatalyst, and the corresponding results are shown in Figure 3. The elements of C, Ag, S, Cd, and O can be clearly observed in the total survey spectrum shown in Figure 3a, where the characteristic binding energy peak of O element comes from the surface adsorbed oxygen of S-C_3_N_4_/AgCdS. Figure 3b–f shows the Cd 3d, Ag 3d, C 1s, S 2p, and N 1s core-level XPS high-resolution spectra, respectively. According to Figure 3b, the characteristic binding energy peaks at 405.18 eV and 411.92 eV correspond to Cd 3d5/2 and Cd 3d2/3, respectively, which are derived from Cd^2+^ in AgCdS [30]. In Figure 3c, the binding energy peaks at 368.22 eV and 374.23 eV are attributed to Ag 3d5/2 and Ag 3d2/3, respectively, which belong to Ag^+^ in AgCdS [31]. The binding energy peaks at 284.78 eV and 288.34 eV in the C 1s core-level spectrum shown in Figure 3d and corresponds to C-C and C-N-C, respectively; the binding energy peak of 286.59 eV comes from the sp3 coordinated carbon species with defects in g-C_3_N_4_ [32,33]. Figure 3e shows the S 2p core-level XPS spectrum, where the characteristic binding energy peaks at 161.53 eV and 162.73 eV are assigned to attributed to S^2−^ in CdS [34]. In Figure 3f, the characteristic binding energy peaks at 398.87 eV, 399.37 eV, and 401.01 eV and corresponds to sp2 nitrogen, sp3 tertiary nitrogen N-(C)_3_, and amino (C-N-H) in the triazine ring (C-N=C), respectively [35,36].

The optical absorption performance of the prepared photocatalysts was further analyzed by testing the UV–visible absorption spectra, and the results are shown in Figure 4. The absorption threshold of pure S-C_3_N_4_ is around 440 nm, while that of pure CdS is around 510 nm. After doping with Ag, the light absorption threshold of AgCdS in the visible light region was slightly expanded, and its optical absorption performance was enhanced in the visible light region of 520–650 nm. The UV–visible absorption spectrum of S-C_3_N_4_/CdS clearly shows two absorption thresholds, which correlate to the characteristic light absorption performance of S-C_3_N_4_ and CdS, respectively. It is undeniable that composite materials of S-C_3_N_4_/CdS and S-C_3_N_4_/AgCdS exhibited better light absorption performance than S-C_3_N_4_, which is mainly attributed to the strong optical absorption performance of CdS and AdCdS. However, S-C_3_N_4_/CdS and S-C_3_N_4_/AgCdS also exhibited weaker optical absorption performance than CdS and AgCdS, which is mainly attributed to the decrease in CdS and AdCdS mass fractions. S-C_3_N_4_/AgCdS exhibited a wider light absorption performance than S-C_3_N_4_/CdS, indicating its higher light absorption and utilization. All prepared photocatalysts exhibited good visible light absorption performance. This will help the absorbance and utilization of visible light to achieve excellent photocatalytic performance.

Figure 5 shows the test results of the photocatalytic degradation performance of the prepared photocatalyst on RhB under visible-light irradiation. As shown in Figure 5a, the prepared photocatalyst exhibited excellent RhB photocatalytic degradation performance. It achieved the removal of over 90% of RhB even under 60 min of light illumination. Among them, S-C_3_N_4_/AgCdS achieved the removal of 99% of RhB under 40 min of light illumination, demonstrating the best photocatalytic RhB degradation performance. To analyze the degree of complete degradation of RhB by the prepared photocatalysts, Figure 5b–f shows the absorption spectra of the degradation solution during the RhB degradation by the prepared photocatalysts. It can be clearly observed that pure CdS and AgCdS exhibit significant shifts in characteristic peaks, indicating that they have not achieved complete degradation of RhB. During the photocatalytic RhB degradation process, S-C_3_N_4_, S-C_3_N_4_/CdS, and S-C_3_N_4_/AgCdS show slight shifts in the characteristic absorption peaks. However, the intensities of all characteristic absorption peaks decrease throughout the entire RhB degradation process, indicating that the complete degradation of RhB can be achieved.

Figure 6 shows the photocatalytic MO degradation performance of the prepared photocatalysts under visible-light irradiation. As shown in Figure 6a, S-C_3_N_4_ and CdS exhibited weak photocatalytic MO degradation performance. AgCdS showed good photocatalytic MO degradation performance in the first 20 min, however, its photocatalytic MO degradation performance weakened after more than 20 min of degradation. Both S-C_3_N_4_/CdS and S-C_3_N_4_/AgCdS exhibited stable and sustained photocatalytic MO degradation performance. S-C_3_N_4_/AgCdS exhibited the best photocatalytic MO degradation performance, achieving a removal of over 95% of MO under 40 min of visible-light irradiation. Figure 6b–f show the absorption spectra of the degradation solution during the MO degradation by different photocatalysts. No other characteristic peaks were observed during the degradation of MO by the prepared photocatalysts, indicating the complete degradation of MO. Based on the results shown in Figure 5 and Figure 6, both S-C_3_N_4_/CdS and S-C_3_N_4_/AgCdS exhibit good photocatalytic RhB and MO degradation performance, while the S-C_3_N_4_/AgCdS photocatalyst exhibits the best photocatalytic RhB and MO degradation performance.

In order to investigate the electron migration ability and the charge carrier recombination rate of S-C_3_N_4_/CdS and S-C_3_N_4_/AgCdS photocatalysts, the EIS spectra and the PL spectra of these two photocatalysts were tested both in the dark and under light illumination, respectively. The corresponding results are shown in Figure 7. Figure 7a shows the EIS spectra of S-C_3_N_4_/CdS and S-C_3_N_4_/AgCdS in the dark and under light illumination, respectively. The smaller the arc radius, the better the electron transfer performance [37]. Under visible-light irradiation, both S-C_3_N_4_/CdS and S-C_3_N_4_/AgCdS exhibited higher electron transfer performance. This is attributed to the formation of more excited photogenerated carriers that flow directionally under the action of heterojunction electric fields, accelerating the separation of photogenerated carriers. S-C_3_N_4_/AgCdS exhibited faster photogenerated carrier migration performance under both in the dark and under light illumination, which is beneficial for the rapid migration of photogenerated carriers to the catalyst surface during the photocatalytic degradation reaction process and thus improves the separation efficiency of the photogenerated charge carriers. Figure 7b shows the PL spectra of S-C_3_N_4_/CdS and S-C_3_N_4_/AgCdS. The weak peak appearing around 440 nm and the typical strong peak appearing around 520 nm are attributed to the characteristic light diffraction peak characteristics of S-C_3_N_4_ and CdS in S-C_3_N_4_/CdS, respectively. The typical peak of S-C_3_N_4_/AgCdS appearing around 520 nm shows a slight right shift, which is attributed to the doping of Ag in CdS. S-C_3_N_4_/CdS exhibited strong fluorescence intensity in the range of 400–700 nm, which can be attributed to the rapid recombination of the photogenerated carriers. In contrast, the PL intensity of S-C_3_N_4_/AgCdS was significantly lower than that of S-C_3_N_4_/CdS, indicating that the recombination rate of the photoinduced electrons and holes generated by S-C_3_N_4_/AgCdS was significantly suppressed. This will definitely achieve effective separation of photogenerated carriers, accelerate the photocatalytic degradation reaction process, and thus possess excellent photocatalytic performance.

The semiconductor types, flat band potentials, and bandgap widths of S-C_3_N_4_, CdS and AgCdS were further explored and the results are shown in Figure 8. Figure 8a–c show the Mott–Schottky curves of the prepared photocatalysts. The tangent slopes of the Mott–Schottky curves of S-C_3_N_4_, CdS, and AgCdS are all positive, indicating that the prepared catalysts are n-type semiconductors. The flat band potential of an n-type semiconductor can be approximately equal to its conduction band (CB) potential [38]. Therefore, the flat band potentials of S-C_3_N_4_, CdS, and AgCdS are roughly determined by the intersection of tangent lines at different frequencies on the horizontal axis. Therefore, the values of the CB potentials of S-C_3_N_4_, CdS, and AgCdS are approximately equal to −1.10 V, −0.45 V, and −0.37 V (vs. Ag/AgCl), respectively. Figure 8d–f show the relations of (αhν)^2^ with E_g_, from which the bandgap widths of S-C_3_N_4_, CdS, and AgCdS can be obtained as 2.84 eV, 2.39 eV, and 2.37 eV, respectively. Based on the equation of E_CB_ = E_VB_ − E_g_ [39], the valence band (VB) potential values of S-C_3_N_4_, CdS, and AgCdS are calculated to be 1.74 V, 1.94 V, and 2.00 V (vs. Ag/AgCl), respectively.

Figure 9 illustrates the energy band structures of CdS, AgCdS, and S-C_3_N_4_, as well as the photocatalytic degradation reaction mechanism of S-C_3_N_4_/AgCdS. As shown in Figure 9a, the VB potential of AgCdS is more positive than that of CdS, which makes the oxidation capability of the photoinduced holes generated by AgCdS stronger than that of the photoinduced holes generated by CdS. Many research results have confirmed that the heterojunction system formed by g-C_3_N_4_/CdS usually follows the photocatalytic reaction mechanism of Z-scheme [40,41,42,43,44]. Compared to CdS, the CB potential of AgCdS is more positive and closer to the valence band of S-C_3_N_4_, making it easier to follow the Z-scheme reaction mechanism in the photocatalytic degradation reaction process. Figure 9b schematically presents the photocatalytic reaction mechanism of the S-C_3_N_4_/AgCdS heterojunction photocatalyst. Under visible-light irradiation, the photoinduced electrons generated by the CB of AgCdS quickly recombine with the photoinduced holes on the VB of S-C_3_N_4_, leaving the photogenerated holes with stronger oxidizing capability in the VB of AgCdS. The photoinduced electrons generated in the CB of S-C_3_N_4_ quickly transfer to the surface of the photocatalyst to participate in the photocatalytic reaction, ultimately achieving the rapid degradation of organic pollutants of RhB and MO.

## 3. Materials and Methods

All reagents used in this experiment were from Shanghai Aladdin Biochemical Technology Co., Ltd. (Shanghai, China), and were not further processed.

### 3.1. Preparation of S-C_3_N_4_ Photocatalyst

S-C_3_N_4_ was prepared in a tube furnace via high-temperature thermal polymerization. Firstly, 3 g of trithiocyanic acid was evenly spread in the porcelain ark. It was then placed in the center of the tube furnace, and was maintained at 600 °C in an air atmosphere for 2 h. The heating and cooling rate of the tube furnace was 5 °C⋅min^−1^. The S-C_3_N_4_ photocatalyst was obtained.

### 3.2. Preparation of CdS and AgCdS Photocatalysts

Pure CdS was prepared through a simple water-bath reaction. 30 mmol of Cd(NO_3_)_2_⋅4H_2_O and 30 mmol of thioacetamide (TAA) were added to 60 mL of deionized water. After adding the stirrer, magnetic stirring was performed to dissolve Cd(NO_3_)_2_⋅4H_2_O and TAA into deionized water. The solution was then placed in a water bath at 60 ℃ for 30 min. The precipitate was collected, washed with deionized water and alcohol three times to remove excess ions, and finally placed in a blast drying oven at 60 °C for 24 h to obtain pure CdS.

### 3.3. Preparation of S-C_3_N_4_/CdS and S-C_3_N_4_/AgCdS Photocatalysts

30 mmol of Cd(NO_3_)_2_⋅4H_2_O and 30 mmol of TAA were added to 60 mL of deionized water. After adding the stirrer, magnetic stirring was performed to dissolve Cd(NO_3_)_2_⋅4H_2_O and TAA into deionized water. A total of 0.2 g of pure S-C_3_N_4_ powder was subsequently added into the solution and it was ultrasonically dispersed for 3 min. This solution was then heated in a water bath at 60 °C for 30 min. The precipitate was collected, washed with deionized water and alcohol three times to remove excess ions, and finally dried at 60 °C for 24 h to obtain S-C_3_N_4_/CdS photocatalyst.

The S-C_3_N_4_/AgCdS photocatalyst was prepared by adding 30 mmol of Cd(NO_3_)_2_⋅4H_2_O, 0.15 mmol of AgNO_3_ and 30 mmol of TAA to 60 mL of deionized water, and repeating the above operation steps to obtain the S-C_3_N_4_/AgCdS photocatalyst.

### 3.4. Characterizations

X-ray diffraction (XRD; TD-3500; Dandong Tongda Technology Co., Ltd. (Dandong, Chian)) was used to identify the crystal structure of the prepared samples. The microstructure of the prepared photocatalyst was observed using a scanning electron microscope (FE-SEM, Ultra 55, Zeiss, Jena, Germany). High-resolution transmission electron microscopy (HRTEM; TecnaiG2F20, FEMompany, Hillsborough, OR, USA) was used to observe the surface microstructure and interface binding information of S-C_3_N_4_/AgCdS. X-ray photoelectron spectroscopy (XPS) was used to analyze the elemental composition and bonding information of composite material S-C_3_N_4_/AgCdS. The optical absorption characteristics of the prepared photocatalysts were measured using a UV–visible near-infrared spectrophotometer (UV vis DRS; UV 3600Plus, Shimadzu, Kyoto, Japan). A fluorescence spectrometer (PL; FluoroMax-4, HORIBA Jobin Yvon, Palaiseau, France) was applied to study the photoluminescence spectra of the prepared photocatalysts.

### 3.5. Photocatalytic Degradation Performance Measurements

50 mg of photocatalyst and 100 mL of RhB solution with the concentration of 10 mg⋅L^−1^ or 100 mL of MO solution with the concentration of 10 mg⋅L^−1^ were used in the photocatalytic degradation experiments. To keep the photocatalyst particles in the adsorption equilibrium state, the mixed solution needed to be stirred continuously in the dark for 30 min. During the photocatalytic degradation process, a 300-W xenon lamp was used as the light source and a UV cut-off filter (λ > 420 nm) was used to generate visible light. The continuous transport of the condensed water was maintained during the entire photocatalytic degradation process to keep the temperature of the solution at 25 °C. After turning on the xenon lamp for illumination, the absorbance and UV–visible absorption spectra of the degradation solution were measured every 10 min using a UV–visible spectrophotometer.

### 3.6. The Preparation and Photoelectrochemical Performance Measurement of the Photoelectrodes

Before preparation of the photoelectrodes, the FTO conductive glass was cleaned, and the specific cleaning steps were as follows. The FTO conductive glass was cut to 20 × 10 mm^2^, was completely immersed in the mixed solution with the volume ratio of deionized water, isopropanol, and anhydrous ethanol of 1:1:1 for a 30 min ultrasonic treatment. Finally, the FTO conductive glass was rinsed with deionized water, and dried for 6 h. An amount of 2 mg of prepared photocatalyst, 10 µL of naphthol, 10 µL of isopropyl alcohol and 150 µL of deionized water were put into an agate mortar, and were carefully ground for 20 min to obtain a uniform dispersion solution. Subsequently, this solution was evenly applied on the conductive surface of the cleaned FTO conductive glass with the coating area controlled at 10 × 10 mm^2^. The photoelectrode could then be obtained after natural drying.

The electrochemical performance of the prepared photocatalysts was tested using a Coaster electrochemical workstation. A conventional three-electrode system was used for the tests: a Pt electrode acted as the counter electrode, an Ag/AgCl electrode was used as the reference electrode, and the prepared photoelectrode acted as the working electrode. An amount of 0.1 M Na_2_SO_4_ solution was used as the electrolyte during the tests. The frequency range of the EIS test was 10–0.01 Hz, with an AC amplitude of 10 mV⋅s^−1^. The measurement was performed both in the dark and under light illumination. For the measurement of the Mott–Schottky curves, the scanning potential range was from −1.0 V to 0.5 V, and the frequency was tested at 500 Hz and at 1000 Hz.

## 4. Conclusions

In the present paper, S-C_3_N_4_/AgCdS Z-scheme heterojunction photocatalyst was successfully prepared. The AgCdS nanoparticles were uniformly loaded onto the surface of S-C_3_N_4_ nanosheets and a good heterojunction structure was formed in the prepared S-C_3_N_4_/AgCdS photocatalyst. The prepared S-C_3_N_4_/AgCdS Z-scheme heterojunction photocatalyst exhibits excellent photocatalytic RhB and MO degradation performance. Through further investigations performed in this work, S-C_3_N_4_/AgCdS more likely follows the photogenerated carriers transport mode of the Z-scheme heterojunction system compared to S-C_3_N_4_/CdS. The Ag doping combined with the Z-scheme heterojunction system enables S-C_3_N_4_/AgCdS to possess a faster photogenerated carrier transport rate and much better separation efficiency of the photogenerated carriers; therefore, the photocatalytic degradation performance of the prepared S-C_3_N_4_/AgCdS photocatalyst is significantly improved.

## Figures and Tables

**Figure 1 molecules-29-01931-f001:**
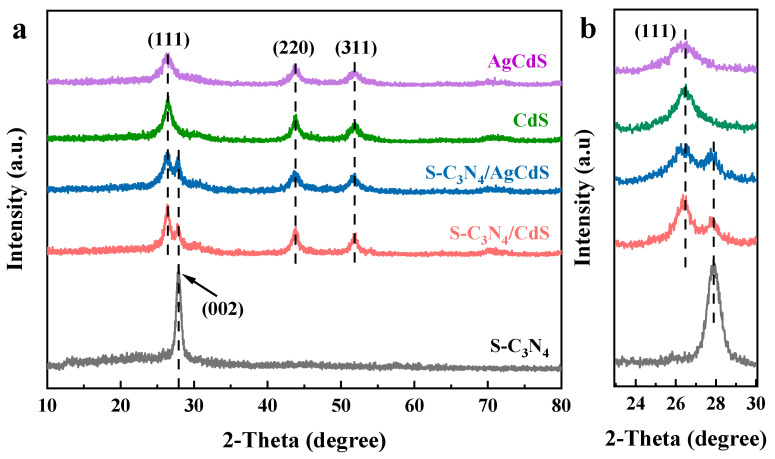
The XRD patterns (**a,b**) of the prepared photocatalysts.

**Figure 2 molecules-29-01931-f002:**
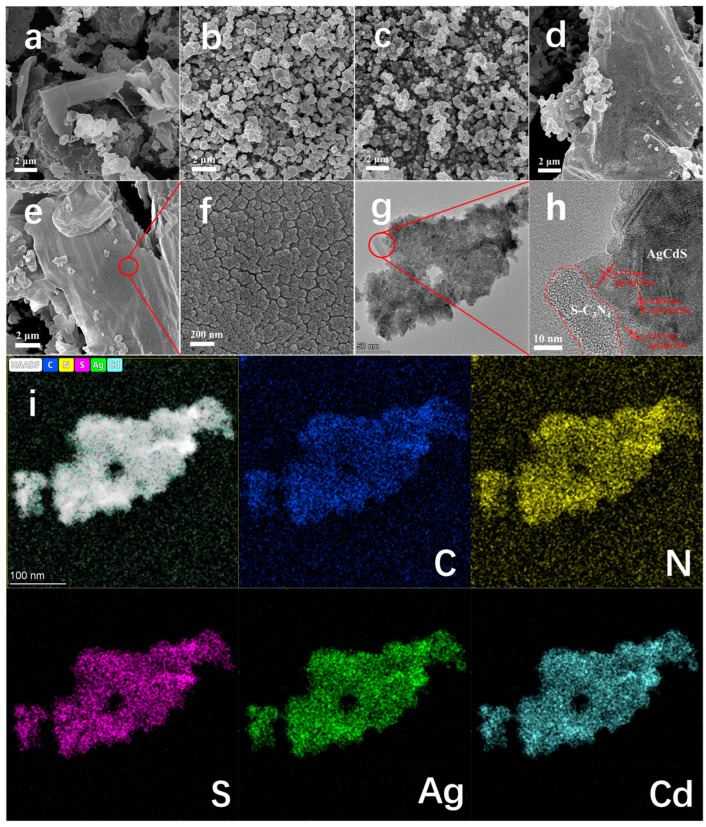
The SEM images of the prepared S-C_3_N_4_ (**a**), CdS (**b**), purAgCdS (**c**), and S-C_3_N_4_/CdS (**d**); the SEM images (**e**,**f**), TEM images (**g**,**h**) and elemental mapping (**i**) of the prepared S-C_3_N_4_/AgCdS.

**Figure 3 molecules-29-01931-f003:**
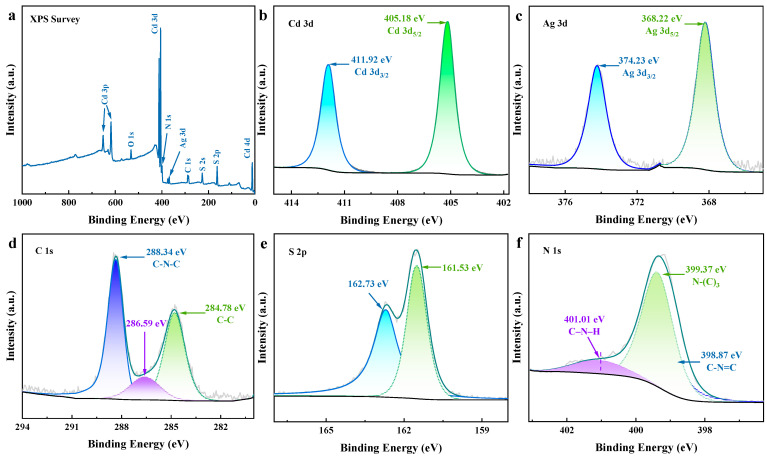
The XPS spectra of the prepared S-C3N4/AgCdS photocatalyst. Total survey spectra (**a**), and Cd3d (**b**), Ag3d (**c**), C1s (**d**), S2p (**e**) and N1s (**f**) XPS core level spectra.

**Figure 4 molecules-29-01931-f004:**
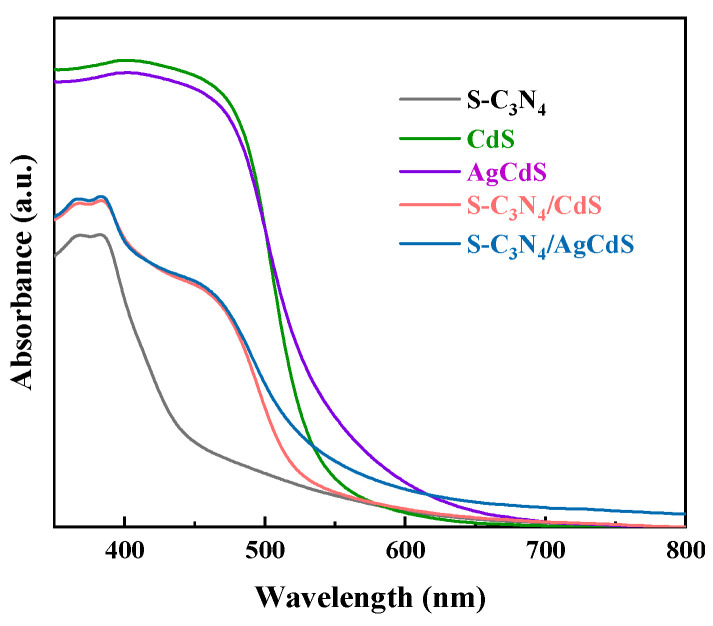
The UV–visible absorption spectra of the prepared photocatalysts.

**Figure 5 molecules-29-01931-f005:**
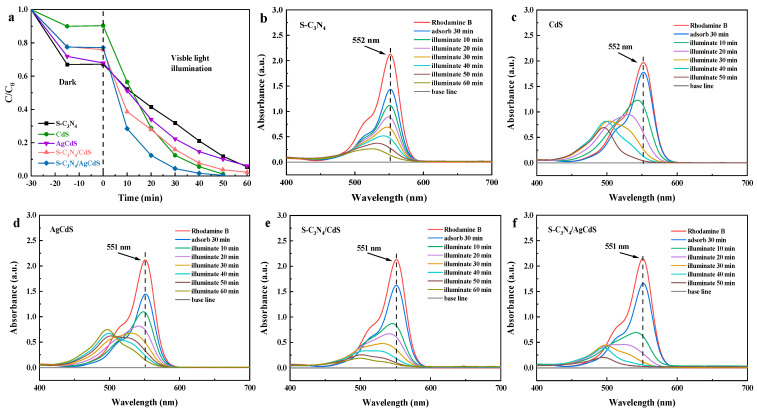
The RhB dye degradation curves (**a**) of the prepared photocatalysts, and the absorption spectra of the degradation solution during the RhB degradation process using different photocatalysts (**b**–**f**).

**Figure 6 molecules-29-01931-f006:**
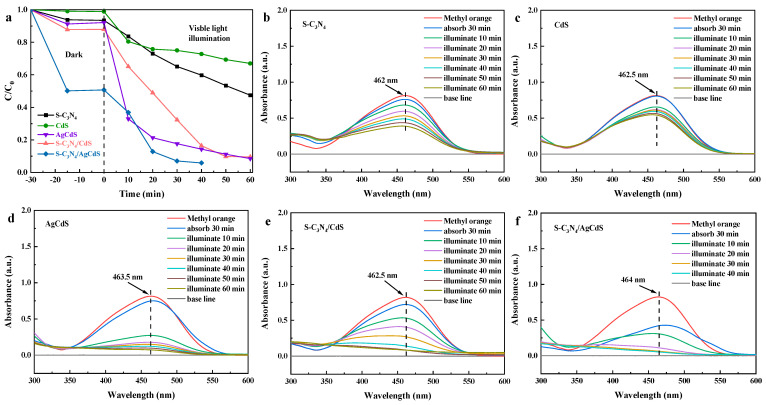
The MO dye degradation curves (**a**) of the prepared photocatalysts, and the absorption spectra of the degradation solution during the MO degradation process using different photocatalysts (**b**–**f**).

**Figure 7 molecules-29-01931-f007:**
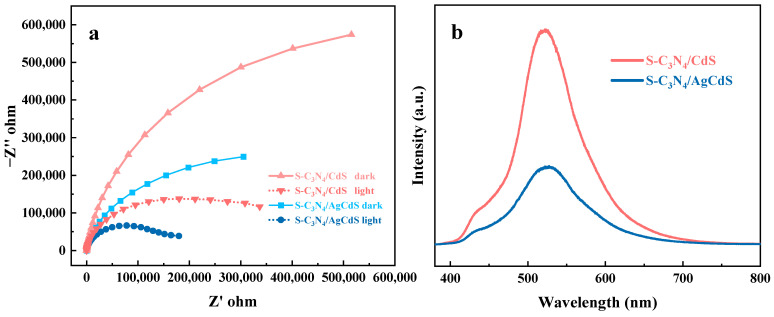
The EIS spectra of the prepared S-C_3_N_4_/CdS and S-C_3_N_4_/AgCdS measured in the dark and under light illumination (**a**), and the PL spectra of the prepared S-C_3_N_4_/CdS and S-C_3_N_4_/AgCdS (**b**).

**Figure 8 molecules-29-01931-f008:**
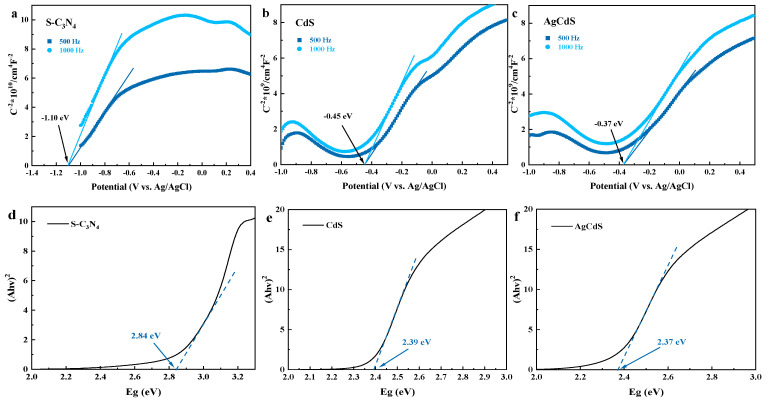
The Mott–Schottky curves (**a**–**c**) and the relations of (αhν)^2^ with E_g_ (**d**–**f**) of the prepared S-C_3_N_4_, CdS, and AgCdS.

**Figure 9 molecules-29-01931-f009:**
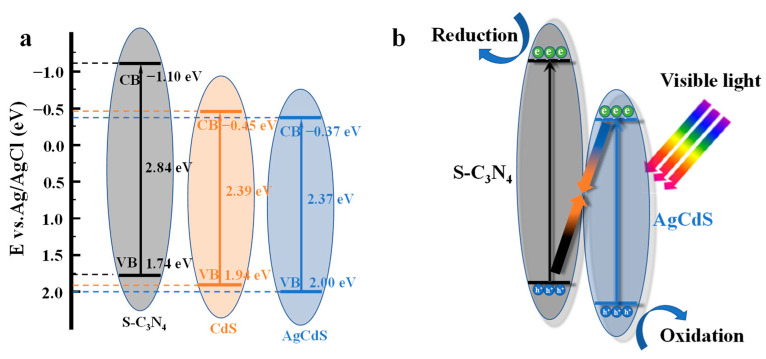
(**a**) Energy band structures of CdS, AgCdS, and S-C_3_N_4_ and (**b**) the proposed photocatalytic reaction mechanism of S-C_3_N_4_/AgCdS.

## Data Availability

Data are contained within the article.

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
