# Peer review of "Preparation of S-C3N4/AgCdS Z-Scheme Heterojunction Photocatalyst and Its Effectively Improved Photocatalytic Performance"

_molecules, 2024, doi:10.3390/molecules29091931_

Round 1

Reviewer 1 Report

Comments and Suggestions for Authors

The authors prepared S-doped graphitic carbon nitride (S-C3N4) by a high-temperature polymerization method. S-C3N4/AgCdS heterojunction photocatalyst was obtained using chemical deposition method by loading Ag-doped CdS nanoparticles (AgCdS NPs) on the surface of S-C3N4. It was found that the S-C3N4/AgCdS heterojunction exhibits excellent photocatalytic degradation performance of Rhodamine B and methyl orange. The doping of Ag in collaboration with the construction of Z-scheme heterojunction system promoted the effective separation and transport of the photogenerated carriers in S-C3N4/AgCdS, significantly accelerated its photocatalytic reaction process, and thus improved its photocatalytic performance. The manuscript is well written and the conclusion can be supported by the data. Thus, the manuscript can be accepted for publication after addressing the following points:

1. The labels for the scale bar in Figure 2 are too small to read. The authors should enlarge the words in the scale bar to show clearer information.

2. For heterojunction, the contents of different materials in the heterojunction must affect the photocatalytic properties. It is suggested to investigate the photocatalytic performances of S-C3N4/AgCdS with different ratios of S-C3N4 and AgCdS to understand the interaction between these two materials.

3. The doping ratio of S in C3N4 and Ag in CdS also affect the photocatalytic properties. Therefore, different ratios of S in C3N4 and Ag in CdS in affecting the photocatalytic performance should also be studied.

4. Some relevant recent papers are recommended to be cited and discussed to broaden the readership: J. Mater. Sci. Technol., 2022, 104, 155-162; J. Mater. Sci. Technol., 2023, 158, 171-179.

Author Response

Reviewer 1

The authors prepared S-doped graphitic carbon nitride (S-C3N4) by a high-temperature polymerization method. S-C3N4/AgCdS heterojunction photocatalyst was obtained using chemical deposition method by loading Ag-doped CdS nanoparticles (AgCdS NPs) on the surface of S-C3N4. It was found that the S-C3N4/AgCdS heterojunction exhibits excellent photocatalytic degradation performance of Rhodamine B and methyl orange. The doping of Ag in collaboration with the construction of Z-scheme heterojunction system promoted the effective separation and transport of the photogenerated carriers in S-C3N4/AgCdS, significantly accelerated its photocatalytic reaction process, and thus improved its photocatalytic performance. The manuscript is well written and the conclusion can be supported by the data. Thus, the manuscript can be accepted for publication after addressing the following points:

Answer: Thank the Reviewer 1 very much! And thank you for giving me the comments to help me improve this manuscript.

  1. The labels for the scale bar in Figure 2 are too small to read. The authors should enlarge the words in the scale bar to show clearer information.

Answer: Thank you for this comment! I have modified the scale bar and clarity in Figure 2 to make to make it clearer to analyze.

Figure 2

  1. For heterojunction, the contents of different materials in the heterojunction must affect the photocatalytic properties. It is suggested to investigate the photocatalytic performances of S-C3N4/AgCdS with different ratios of S-C3N4and AgCdS to understand the interaction between these two materials.

Answer: Thank you for this comment! For heterojunction, the contents of different materials in the heterojunction must affect the photocatalytic properties. This manuscript mainly reflects the significantly improved photocatalytic performance of Z-scheme heterojunctions. In future research, heterojunction photocatalysts with different ratios will be further investigated, and we believe they will soon be presented to readers. Thank you very much for your comment.

  1. The doping ratio of S in C3N4and Ag in CdS also affect the photocatalytic properties. Therefore, different ratios of S in C3N4and Ag in CdS in affecting the photocatalytic performance should also be studied.

Answer: Thank you for your comment! Like the above-mentioned comment, this manuscript mainly reflects the significantly improved photocatalytic performance of Z-scheme heterojunctions. In fact, we have already conducted preliminary exploration of some of the content. In future research, heterojunction photocatalysts with different doping ratio of S in C3N4 and Ag in CdS will be further investigated, and we believe they will soon be presented to readers. Thank you very much for your comment.

  1. Some relevant recent papers are recommended to be cited and discussed to broaden the readership: J. Mater. Sci. Technol., 2022, 104, 155-162; J. Mater. Sci. Technol., 2023, 158, 171-179.

Answer: Thank you for your comment! The provided literatures are very helpful for this article, and I have already made relevant literature supplementation in this paper.

Reviewer 2 Report

Comments and Suggestions for Authors

In this article Y. Lin et.al., demonstrated the study of photocatalytic activity based on Z-Scheme S-C3N4/AgCdS heterojunction photocatalyst. The photocatalyst exhibits excellent photocatalytic degradation performance of Rhodamine B and methyl orange. The manuscript was well written and the concept theme is good and have done an extensive analysis with a suitable band diagram. The results presented in the study are good and can be likely to attract the readers in in the field of photocatalytic applications. However, there are some minor issues that needs to be clarified before processing for the further step toward publication, and here are some of my comments as given below.

1. Please describe the significance of Z-scheme of your structure over the type-II heterostructure.

2. Does doping of S atoms/Ag atoms in to the g-C3N4 and CdS make any changes in the g-C3N4 and CdS matrix like optoelectrical and recombination centers. Please give detail account that makes much attentions to your work novelty in revised manuscript.

3. How did the authors confess the charge transfer between the Z-Scheme heterostructure material in revised manuscript?

4. From UV visible please explain the reason in the increase in absorbance level of the hybrid composite in revised manuscript.

5. It could be better to the readers understanding if authors provide the pristine g-C3N4 PL spectra for references. Also, please describe the peak positions and their significance.

6. Some of the latest reports are suggested to cite at the heterostructure, photocatalytic, 2D materials, and related discussions, such as Nano Research 16 (5), 7682-7695. Chemosphere 311 (2023): 137000.

Author Response

Reviewer 2

In this article Y. Lin et.al., demonstrated the study of photocatalytic activity based on Z-Scheme S-C3N4/AgCdS heterojunction photocatalyst. The photocatalyst exhibits excellent photocatalytic degradation performance of Rhodamine B and methyl orange. The manuscript was well written and the concept theme is good and have done an extensive analysis with a suitable band diagram. The results presented in the study are good and can be likely to attract the readers in in the field of photocatalytic applications. However, there are some minor issues that needs to be clarified before processing for the further step toward publication, and here are some of my comments as given below.

 Answer: Thank the Reviewer 2 very much! And thank you for giving me the comments to help me improve this manuscript.

  1. Please describe the significance of Z-scheme of your structure over the type-II heterostructure.

Answer: Thank you for your comment! The Z-scheme photocatalysts can utilize the conduction band with stronger reduction ability and the valence band with stronger oxidation ability, which means they have stronger oxidation-reduction ability. Although type-II photocatalysts integrate photogenerated carriers generated by different photocatalysts, their oxidation-reduction ability is significantly insufficient. Therefore, Z-scheme photocatalysts are more likely to achieve rapid removal of pollutants. I have added the relevant description to the manuscript. Secondly, the mechanism analysis in Figure 9 also expresses the same viewpoint. Thank you for your comment.

  1. Does doping of S atoms/Ag atoms in to the g-C3N4 and CdS make any changes in the g-C3N4 and CdS matrix like optoelectrical and recombination centers. Please give detail account that makes much attentions to your work novelty in revised manuscript.

Answer: Thank you for your comment! S-C3N4 was directly prepared in a tube furnace by high temperature thermal polymerization. Previous reports have confirmed its superior performance compared to g-C3N4. For CdS and AgCdS photocatalyst:Firstly, the XRD results have confirmed the successful doping of Ag into CdS in Figure 1. Secondly, the photocatalytic degradation results also demonstrated the advantages of AgCdS in Figures 5 and 6. Thirdly, the research focus of the manuscript is on Z-Scheme S-C3N4/AgCdS heterojunction photocatalyst. I compared the photogenerated carrier migration performance and secondary recombination ability of S-C3N4/AgCdS and S-C3N4/CdS, and the results are shown in Figure 7. S-C3N4/AgCdS exhibits faster photogenerated carrier migration performance under both in the dark and under light illumination, which also have significantly suppressed secondary electron recombination ability. The main innovation of the manuscript lies in the use of doped photocatalysts to establish an effective heterojunction system and significantly improve its photocatalytic performance. In the process of establishing heterojunctions, various modification methods of photocatalysts such as doping, morphology control, and heterojunction establishment were fully utilized in this work. The relevant content has been added to the manuscript.

  1. How did the authors confess the charge transfer between the Z-Scheme heterostructure material in revised manuscript?

Answer: Thank you for your comment! A lot of research results had confirmed that the heterojunction system formed by g-C3N4/CdS usually follows the photocatalytic reaction mechanism of Z-scheme (Applied Catalysis B: Environmental, 2015, 168: 465-471, Applied Surface Science, 2019, 478: 1056-1064, Chemical Engineering Journal, 2017, 317: 913-924.). Furthermore, Figure 9 illustrates the energy band structures of CdS, AgCdS and S-C3N4. Compared to CdS, the CB potential of AgCdS is more positive and closer to the valence band of S-C3N4, making it be easier to follow the Z-scheme reaction mechanism in the photocatalytic degradation reaction process. In addition, the CB potential of CdS is more negative than that of AgCdS, and the reduction ability of electrons participating in the reaction is stronger. If the transport of photogenerated carriers in composite photocatalysts follows the type-II reaction mechanism, this obviously contradicts the results of this study. Therefore, the S-C3N4/AgCdS photocatalyst prepared in this work follows the Z-scheme photocatalytic reaction mechanism.

  1. From UV visible please explain the reason in the increase in absorbance level of the hybrid composite in revised manuscript.

Answer: Thank you for your comment! As can been found from Figure 4, the UV visible absorption spectrum of S-C3N4/CdS clearly shows two absorption thresholds, which attribute to the characteristic light absorption performance of S-C3N4 and CdS, respectively. It is undeniable that composite materials of S-C3N4/CdS and S-C3N4/AgCdS exhibit better light absorption performance than S-C3N4, which is mainly attributed to the strong optical absorption performance of CdS and AdCdS. While S-C3N4/CdS and S-C3N4/AgCdS also exhibiting weaker optical absorption performance than CdS and AgCdS, which is mainly attributed to the decrease in CdS and AdCdS mass fractions. S-C3N4/AgCdS exhibits a wider light absorption performance than S-C3N4/CdS, indicating its higher light absorption and utilization.

  1. It could be better to the readers understanding if authors provide the pristine g-C3N4 PL spectra for references. Also, please describe the peak positions and their significance.

Answer: Thank you for your comment! Figure 7b shows the PL spectra of S-C3N4/CdS and S-C3N4/AgCdS. The PL spectra of S-C3N4 and CdS are not provided here mainly to compare the secondary electron recombination ability of the two composite materials, in order to verify the advantages of constructing heterojunction system with doped photocatalyst. Therefore, we did not provide the spectra of S-C3N4 and CdS. Figure 7b shows the PL spectra of S-C3N4/CdS and S-C3N4/AgCdS. The weak peak appearing around 440 nm and the typical strong peak appearing around 520 nm are attributed to the characteristic light diffraction peak characteristics of S-C3N4 and CdS in S-C3N4/CdS, respectively. The typical peak of S-C3N4/AgCdS appearing around 520 nm shows a slight right shift, which is attributed to the doping of Ag in CdS. S-C3N4/CdS exhibits strong fluorescence intensity in the range of 400-700 nm, which can be attributed to the rapid recombination of the photogenerated carriers. In contrast, the PL intensity of S-C3N4/AgCdS is significantly lower than that of S-C3N4/CdS, indicating that the recombination rate of the photoinduced electrons and holes generated by S-C3N4/AgCdS is significantly suppressed.

  1. Some of the latest reports are suggested to cite at the heterostructure, photocatalytic, 2D materials, and related discussions, such as Nano Research 16 (5), 7682-7695. Chemosphere 311 (2023): 137000.

Answer: Thank you for your comment! The provided literatures are very helpful for this article, and I have already made relevant literature supplementation in this paper.

Round 2

Reviewer 1 Report

Comments and Suggestions for Authors

The authors have addressed all comments from reviewers. The current version of the manuscript can be accepted for publication.